# The Therapeutic Effect of Monopolar Radiofrequency Therapy on Urinary Symptoms and Sexual Function

**DOI:** 10.3390/biomedicines12102288

**Published:** 2024-10-09

**Authors:** Cheng-Yu Long, Chieh-Yu Chang, I-Chieh Sung, Zi-Xi Loo, Kun-Ling Lin

**Affiliations:** 1Department of Obstetrics and Gynecology, Kaohsiung Medical University Hospital, Kaohsiung Medical University, Kaohsiung 80708, Taiwan; starrynight278@gmail.com (I.-C.S.); 1030394@kmuh.org.tw (Z.-X.L.); 960233@kmuh.org.tw (K.-L.L.); 2Department of Obstetrics and Gynecology, Kaohsiung Municipal Siao-Gang Hospital, Kaohsiung Medical University, Kaohsiung 80708, Taiwan; judychang.9236@gmail.com

**Keywords:** stress urinary incontinence (SUI), radiofrequency (RF) therapy, female sexual function, Viveve^®^ System, urodynamic studies, non-surgical treatment

## Abstract

**Objectives:** Stress urinary incontinence (SUI) negatively affects the quality of life and sexual function in women. This study aimed to evaluate the efficacy of radiofrequency (RF) therapy in reducing SUI symptoms and its impact on sexual function. **Methods:** Thirty-four women with SUI were enrolled and underwent a single RF treatment session using the Viveve^®^ System (Viveve Medical Inc., USA) with parameters of 90 J/cm^2^ and 220 pulses per hour. Assessments at baseline and 6 months post treatment included perineal ultrasound and personal interviews to evaluate lower urinary tract symptoms and sexual function. Urodynamic studies, voiding diaries, and questionnaires such as the Female Sexual Function Index (FSFI), Overactive Bladder Symptom Score (OABSS), Urogenital Distress Inventory-6 (UDI-6), Incontinence Impact Questionnaire-7 (IIQ-7), and International Consultation on Incontinence Questionnaire—Short Form (ICIQ-SF) measured outcomes. **Results:** RF therapy significantly improved sexual function, with higher FSFI scores in all domains except pain at 6 months. SUI symptoms were significantly reduced, as indicated by improved scores on OABSS, UDI-6, IIQ-7, and ICIQ-SF, alongside better voiding diary results. Anatomical changes included reduced bladder neck mobility, decreased vaginal width, and a reduced rotation angle of the proximal urethra. **Conclusions:** RF therapy is effective and safe for treating mild to moderate SUI and enhances sexual function, potentially due to changes in vaginal topography. These results suggest RF therapy as a viable non-surgical option for managing SUI and improving sexual health.

## 1. Introduction

Stress urinary incontinence (SUI) is a prevalent lower urinary tract disorder characterized by involuntary urine leakage during activities that increase intra-abdominal pressure, such as exercise, laughing, or coughing [1]. From 2005 to 2016, urinary incontinence affected approximately 53% of women, with 26% attributed to SUI [2]. In the U.S., sixteen billion dollars are spent annually on treating urinary incontinence, with thirteen billion specifically allocated to SUI [3]. Similarly, annual medical costs in Taiwan doubled or tripled from 1997 to 2011, highlighting the rising prevalence of SUI despite advancements in medical care [4].

SUI is most commonly seen in women, and its prevalence increases with age, particularly among postpartum and postmenopausal women. Various risk factors contribute to SUI, including obesity, childbirth-related pelvic floor trauma, hormonal changes (e.g., decreased estrogen), genetic predisposition, and lifestyle factors such as high-impact physical activity [5]. Chronic conditions that elevate intra-abdominal pressure, like persistent coughing or constipation, further exacerbate the risk of developing SUI. Additionally, race and ethnicity appear to play a role, with some studies indicating a higher prevalence among Caucasian women compared to African American and Asian populations [6].

The impact of SUI extends beyond physical symptoms, with significant implications for mental health and sexual function. Untreated SUI can lead to worsening urinary symptoms, social withdrawal, and impaired intimacy due to embarrassment or anxiety related to urinary leakage [7]. While effective non-surgical treatments for moderate SUI include pelvic floor muscle training, biofeedback, and electrical stimulation, surgical interventions such as mid-urethral sling implantation are recommended for severe cases [8]. However, these surgical and medical treatments come with risks, including voiding dysfunction, urinary tract infections, and post-operative complications [8,9]. As such, there is an urgent need for alternative, minimally invasive therapeutic strategies for SUI that present fewer risks and side effects.

In recent years, non-ablative energy-based therapies like fractional lasers, radiofrequency (RF), and intense pulsed light (IPL), initially used for skin rejuvenation, have been adapted for vulvovaginal rejuvenation [10,11]. The pathophysiology of SUI is often linked to the relaxation or weakening of the urethral sphincter and pelvic floor muscles [12]. RF therapy, which emits focused electromagnetic waves to heat tissues, is of particular interest as it induces collagen folding and elastin regeneration, enhancing tissue elasticity and restoring urethral support [13,14]. Histological studies have demonstrated the beneficial effects of RF on vulvovaginal tissue, supporting its potential in improving SUI symptoms [15,16].

Pelvic floor ultrasound is a valuable tool for evaluating the anatomical and functional aspects of the lower urinary tract. It provides objective measurements of urethral and bladder neck mobility, urethral angles, and pelvic organ support, allowing for a more precise assessment of SUI [17,18]. Despite the promising use of RF therapy for SUI, no studies have yet utilized pelvic floor ultrasound to objectively validate its effects on urethral angles or pelvic floor structure.

Women with SUI frequently experience sexual dysfunction, which significantly impacts their quality of life. Urine leakage during sexual activity can lead to anxiety, reduced sexual desire, and avoidance of intimacy. The severity of pelvic organ prolapses, often associated with SUI, is directly correlated with sexual dysfunction [19,20]. The psychosocial burden of urinary incontinence, especially during midlife, often results in feelings of inadequacy and strained intimate relationships [21]. Addressing both urinary and sexual symptoms is therefore crucial in comprehensive SUI management.

Recent research suggests that the urinary and vaginal microbiota might play a role in the pathophysiology of SUI. Disruptions in the balance of these microbial communities, known as dysbiosis, have been associated with various urological conditions, such as urinary tract infections and overactive bladder syndrome [22,23]. While the connection between SUI and microbiota remains largely unexplored, altered microbiota could potentially influence pelvic floor health, inflammation, and tissue integrity, thus contributing to SUI symptoms [24]. More research is warranted to investigate this relationship and its implications for treatment.

## 2. Materials and Methods

### 2.1. Study Design

This study was conducted at the Department of Obstetrics and Gynecology, Kaohsiung Medical University Hospital, from March 2019 to February 2021. The study initially involved 39 patients with mild to moderate SUI. The inclusion criteria were (1) pre-surgical condition for SUI, (2) age > 20 years, (3) sexual activity within the past three months, and (4) postpartum period > six weeks. The exclusion criteria included (1) vaginal bleeding, (2) malignancies, (3) urinary infections, (4) pelvic organ prolapse, (5) pregnancy, (6) vaginitis or other infections, (7) implanted medical devices, (8) genital fistulas, and (9) vulvodynia. Five patients withdrew due to reasons including loss to follow-up and personal withdrawal of consent, leaving 34 participants for final analysis.

### 2.2. Intervention Procedure

The intervention involved a single session of vaginal monopolar radiofrequency therapy using the Viveve^®^ System (Viveve Medical Inc., Englewood, CO, USA) with an energy output of 90 J/cm^2^ over one hour (220 pulses). Baseline and six-month post-treatment assessments were conducted.

### 2.3. Outcome Measurements and Assessments

#### 2.3.1. Questionnaire-Based Assessments:

Assessment tools included the Vaginal Laxity Questionnaire (VLQ) [25], Overactive Bladder Symptom Score (OABSS) [26], Urogenital Distress Inventory 6 (UDI-6) [27], Incontinence Impact Questionnaire 7 (IIQ-7) [27], Incontinence Questionnaire—Short Form (ICIQ-SF) [28], and Female Sexual Function Index (FSFI) [29]. A pad test measured urine leakage, with a >1 g increase over one hour indicating a positive result [30].

#### 2.3.2. Urodynamic Assessments:

Urodynamic studies, including uroflowmetry, cystometry, and urethral pressure profilometry, were performed using a 6-channel urodynamic monitor (MMS; UD2000, Enschede, The Netherlands). Positive detrusor overactivity was noted based on uninhibited contractions during cystometry.

#### 2.3.3. Voiding Diary

Participants maintained a voiding diary to record the frequency, volume, and episodes of incontinence over a 24 h period. These diaries were reviewed during follow-up assessments to provide further insight into the treatment’s impact on SUI symptoms.

### 2.4. Imaging Assessments

Trans-perineal ultrasound measured bladder neck mobility, vaginal area, and proximal urethral rotation angle using a Volusion General Electric Sonography 730 Expert device (GE Healthcare, Chicago, IL, USA) with a 3.5 MHz curved linear array transducer. Measurements were taken at baseline and six months post treatment under both rest and strain conditions.

### 2.5. Ethical Approval and Consent

The study protocol was sanctioned by the Ethics Committee of Kaohsiung Medical University Chung-Ho Memorial Hospital, adhering to the principles of the Declaration of Helsinki. Informed consent was obtained from all participants before treatment.

## 3. Results

### 3.1. Participant Characteristics and Baseline SUI Severity

The study included 34 participants, with a mean age of 43.8 ± 8.8 years and a mean BMI of 22.7 ± 3.5 kg/m^2^. The distribution of SUI severity assessed by the ICIQ was 17.7% mild, 64.7% moderate, and 17.7% severe before treatment. After six months of treatment, 76.5% (26 out of 34) of participants showed significant improvement in SUI symptoms (Table 1).

### 3.2. Improvements in Sexual Function

Sexual function was evaluated at baseline and six months post-treatment using the FSFI (Table 2). Significant improvements were observed across various domains. The mean score for sexual desire increased from 3.0 ± 0.8 at baseline to 3.5 ± 0.9 post-treatment (*p* = 0.002). Sexual arousal also improved, with the mean score rising from 3.1 ± 0.8 to 3.7 ± 0.9 (*p* = 0.001). Participants reported an increase in orgasm scores from 3.5 ± 1.3 to 4.0 ± 1.2 (*p* = 0.010), while sexual satisfaction scores rose from 3.9 ± 1.3 to 4.4 ± 1.1 (*p* = 0.015). Overall, the total FSFI score increased from 22.2 ± 5.9 to 25.6 ± 5.0 (*p* = 0.003), with 70.6% of participants (24 out of 34) showing improved total scores.

### 3.3. Improvements in Urinary Symptoms and Quality of Life

Questionnaire assessments showed significant improvements in urinary distress and quality of life at Table 3. The OABSS decreased from 5.3 ± 3.3 at baseline to 3.3 ± 2.2 post treatment (*p* = 0.02). The UDI-6 scores improved from 29.6 ± 13.9 to 17.7 ± 11.2 (*p* < 0.01), and the IIQ-7 scores decreased from 22.1 ± 16.9 to 9.9 ± 13.0 (*p* < 0.01). The ICIQ-SF scores improved from 8.7 ± 3.4 to 5.9 ± 3.7 (*p* < 0.01). Additionally, the VLQ scores increased from 3.15 ± 1.0 to 4.1 ± 1.2 (*p* < 0.01), with 67.7% of participants (23 out of 34) reporting higher VLQ scores.

### 3.4. Urodynamic Changes

Urodynamic assessments indicated a few significant changes following treatment (Table 4). The mean pad test result decreased from 12.8 ± 19.6 g to 5.0 ± 13.6 g (*p* = 0.013). Detrusor pressure at peak flow increased from 10.0 ± 28.5 cm H_2_O to 22.7 ± 14.8 cm H_2_O (*p* = 0.016). Other urodynamic parameters, such as maximum flow rate (Qmax), residual urine (RU), bladder volume at first desire to void (Vfst), maximum cystometric capacity (MCC), maximum urethral closure pressure (MUCP), functional urethral length (FUL), and urethral closure pressure area (UCA), did not show statistically significant changes.

### 3.5. Voiding Diary Outcomes

The voiding diary results revealed notable improvements in urinary symptoms (Table 5). The frequency of urination per 24 h decreased from 8.1 ± 2.8 times to 7.2 ± 2.1 times (*p* = 0.034), and episodes of urge incontinence per 24 h significantly decreased from 2.0 ± 1.9 to 0.9 ± 1.4 (*p* = 0.001). Other parameters, such as voided urine volume per time, maximum urine volume, and average nocturia per 24 h, did not show significant changes.

### 3.6. Changes in Vaginal and Urethral Topography

Changes in vaginal and urethral topography were also observed post treatment in Table 6. Bladder neck mobility decreased from 1.6 ± 0.3 to 1.3 ± 0.2 (*p* = 0.003). Vaginal width at Valsalva and area at both resting and Valsalva showed significant reductions. At rest, vaginal width decreased from 3.1 ± 0.4 cm to 2.9 ± 0.3 cm (*p* = 0.080) and vaginal area decreased from 3.1 ± 0.5 cm^2^ to 2.8 ± 0.5 cm^2^ (*p* = 0.018). During the Valsalva maneuver, vaginal width decreased from 3.5 ± 1.1 cm to 2.9 ± 0.8 cm (*p* = 0.004), and vaginal area decreased from 4.0 ± 1.0 cm^2^ to 3.1 ± 0.9 cm^2^ (*p* = 0.001). The proximal urethral rotation angle reduced from 15.3 ± 5.0 degrees to 11.6 ± 3.1 degrees (*p* = 0.009).

## 4. Discussion

RF energy, functioning within a frequency range of 20 kHz to 300 GHz [31], has been employed in medical treatments for over 125 years, including diathermy, hyperthermia treatments, electrosurgical scalpels, and radiofrequency ablation [32,33]. Another key application is magnetic resonance imaging (MRI), which utilizes RF to generate body images. Our current research focuses on assessing RF’s therapeutic effects on SUI and sexual function over 6 months. This study measures standard questionnaire outcomes related to SUI symptoms and sexual function, as well as bladder neck mobility and urethral rotation angles, to accurately gauge SUI conditions and potentially prevent unnecessary surgical interventions. The increasing demand for non-invasive methods like RF and laser treatments for various vaginal issues has been noted [34,35].

Studies have highlighted the beneficial impacts of laser treatments on sexual function. Eder et al., 2019 reported that 15 participants receiving two fractional CO_2_ laser treatments—one at baseline and a maintenance treatment at either 12 or 15 months—saw significant improvements in total FSFI scores at follow-ups of 12, 15, and 18 months, with scores increasing from 16.2 ± 7.9 to 24.4 ± 6.9, 22.2 ± 6.7, and 25.8 ± 6.6, respectively. This underscores the laser’s efficacy in treating sexual dysfunction caused by post-menopausal vaginal atrophy over a long-term period [36]. However, a study by Lou et al., 2022, compared vaginal fractional CO_2_ laser therapy to Kegel exercises for female sexual dysfunction and found no significant difference in FSFI total scores at a 12-month follow-up, except in the lubrication category [37].

The effects of Er: YAG vaginal laser treatment on women’s sexual dysfunction, specifically in patients with SUI, were explored in a prior study, which observed a significant improvement in the overall FSFI scores, increasing from 22.2 ± 6.2 to 25.6 ± 4.5 after six months of treatment. Although the increase in the sexual desire domain was modest (from 2.8 ± 1.2 to 3.0 ± 1.0, *p* = 0.07), the results suggest potential benefits of this treatment. Additionally, in a comparative assessment, the use of RF treatment significantly boosted the FSFI total score from 22.2 ± 5.9 to 25.6 ± 5.0 within six months, enhancing many domains in the FSFI [38]. These findings indicate that RF treatment may offer more rapid efficacy in improving sexual dysfunction than the Er: YAG laser. However, further research with a longer follow-up period is necessary to confirm the durability and full scope of these therapeutic effects.

Recent studies have explored non-ablative laser therapy for treating SUI. Nalewczynska et al. (2022) demonstrated the safety and effectiveness of pixel CO_2_ laser, noting slight symptom improvements and suggesting the need for maintenance treatments within 6 to 12 months [39].

The efficacy of two types of laser therapy—Er: YAG and pixel CO_2_—on SUI symptoms was investigated, with data collected over a 6-month period. The Er: YAG laser showed significant improvements in the UDI-6 and IIQ-7 questionnaires (*p* = 0.006; *p* = 0.005), which assess the distress and impact of urinary incontinence. Additionally, the OABSS and POPDI-7 scores, which reflect discomfort from pelvic organ prolapse and overactive bladder syndrome, also improved significantly (*p* = 0.001; *p* = 0.037) [40]. Conversely, three treatments with the pixel CO_2_ laser yielded mixed results, with significant improvements in UDI-6 and IIQ-7 (*p* = 0.012; *p* = 0.049) but no significant change in OABSS scores (*p* = 0.481) [41].

In contrast to laser therapy, RF treatment demonstrated more rapid improvements in SUI symptoms. Within six months, significant reductions were observed in UDI-6, IIQ-7 (*p* < 0.01), and OABSS scores (*p* = 0.02). These results suggest that RF may be more effective than laser therapy in the short term for alleviating SUI symptoms.

Lin et al. (2017) confirmed that laser therapy improves overactive bladder symptoms and urodynamic parameters, although some benefits did not persist beyond a year [42]. Blaganje et al. (2018) found that Er: YAG laser treatment significantly enhanced duration and maximum pressure during pelvic exercises, but not average pressure [43]. Alcalay et al. (2021) observed a notable reduction in the 1 h pad test with the pixel CO_2_ laser, with 41.4% of patients showing no SUI at 6 months [44], yet our results showed no significant urodynamic changes at 6 months. In contrast, RF treatment significantly improved both the 1 h pad test and detrusor pressure, indicating its effectiveness (*p* < 0.05). The findings from the voiding bladder diary corroborate our earlier observations. Our current data show that RF treatment significantly reduced the frequency of urination per 24 h (*p* = 0.034) and the incidents of urge incontinence within the same timeframe (*p* = 0.001 *).

Assessments of bladder neck mobility via perineal ultrasound have become crucial in validating SUI status. The findings from the Er: YAG and pixel CO_2_ laser treatments showed significant decreases in bladder neck mobility and middle urethral area [38,41]. RF treatment further decreased bladder neck mobility (*p* = 0.003) and significantly reduced vaginal width at Valsalva and area under both rest and straining conditions (*p* < 0.05), as well as the proximal urethral rotation angle (*p* = 0.009). These changes suggest a strong correlation between RF therapy and the recovery of SUI symptoms. During the treatment, none of these cases reported obvious side effects.

The study faces limitations that could impact the scope and interpretation of the results. The small sample size restricts the generalizability of the findings and diminishes the robustness of statistical conclusions. A brief follow-up period of only 6 months may not adequately capture the long-term effects and any adverse outcomes associated with the RF and laser treatments. Another limitation of this study is that it did not specifically assess sexual distress. The focus of the study was on evaluating improvements in sexual function using the Female Sexual Function Index (FSFI), without the inclusion of distress-related questionnaires. Although sexual function showed significant improvement, future studies could include assessments of sexual distress to explore this aspect more comprehensively in patients with SUI.

## 5. Conclusions

Our study indicates that a single vaginal RF treatment can markedly improve SUI symptoms, reflected in both questionnaire responses and perineal ultrasound measurements. The RF treatment also demonstrated significant improvements across various FSFI indexes and total scores. Given these promising results, additional randomized trials are recommended to further assess the safety and long-term efficacy of RF therapy in treating women with SUI and potentially other related conditions.

## Figures and Tables

**Table 1 biomedicines-12-02288-t001:** The clinical background of the participants. Data are given as mean ± standard deviation or n (%).

	Pre-Treatment (n = 34)	Post-Treatment (n = 34)
Mean age (years)	43.8 ± 8.8	
Mean BMI (kg/m^2^)	22.7 ± 3.5	
SUI grade by ICIQ		
Mild	6 (17.7)	
Moderate	22 (64.7)	
Severe	6 (17.7)	
Very severe	0	
Efficacy for SUI		26/34 (76.5%)
Follow-up (months)		6 months

BMI, body mass index; SUI, stress urinary incontinence; ICIQ, International Consultation on Incontinence Questionnaire. *p* < 0.05, Student’s *t*-test.

**Table 2 biomedicines-12-02288-t002:** Changes in sexual function before and six months post treatment. Data are given as mean ± standard deviation or n (%).

n = 34	Baseline	6 Months Post Treatment	*p* Value *
Desire (1, 2)	3.0 ± 0.8	3.5 ± 0.9	0.002 *
Arousal (3–6)	3.1 ± 0.8	3.7 ± 0.9	0.001 *
Lubrication (7–10)	4.2 ± 1.4	4.7 ± 1.0	0.058
Orgasm (11–13)	3.5 ± 1.3	4.0 ± 1.2	0.010 *
Satisfaction (14–16)	3.9 ± 1.3	4.4 ± 1.1	0.015 *
Pain (17–19)	4.6 ± 1.5	5.1 ± 1.2	0.089
FSFI total scores	22.2 ± 5.9	25.6 ± 5.0	0.003 *
Rate of improved total scores		24/34 (70.6%)	

* Statistical significance; Paired *t*-test.

**Table 3 biomedicines-12-02288-t003:** Questionnaire results before and six months post treatment. Data are given as mean ± standard deviation or n (%).

n = 34	Baseline	6 Months Post Treatment	*p* Value *
OABSS	5.3 ± 3.3	3.3 ± 2.2	0.02 *
UDI-6	29.6 ± 13.9	17.7 ± 11.2	<0.01 *
IIQ-7	22.1 ± 16.9	9.9 ± 13.0	<0.01 *
ICIQ-SF	8.7 ± 3.4	5.9 ± 3.7	<0.01 *
VLQ	3.15 ± 1.0	4.1 ± 1.2	<0.01
Rate of higher VLQ		23/34 (67.7%)	

VLQ, Vaginal Laxity Questionnaire; OABSS, Overactive Bladder Symptom Score; UDI-6, Urinary Distress Index; IIQ-7, Incontinence Impact Questionnaire; ICIQ-SF, International Consultation on Incontinence Questionnaire—Short Form. Values are expressed as mean ± standard deviation or numbers * Statistical significance; Paired *t*-test.

**Table 4 biomedicines-12-02288-t004:** Urodynamic changes at baseline and six months after treatment. Data are given as mean ± standard deviation.

n = 34	Baseline	6 Months Post-Treatment	*p* Value *
Pad test	12.8 ± 19.6	5.0 ± 13.6	0.013 *
Vaginal Pressure (cm H_2_O)	53.9 ± 16.4	53.3 ± 25.9	0.880
Qmax (mL/s)	26.3 ± 9.7	24.1 ± 11.4	0.214
RU (mL)	40.3 ± 42.5	49.8 ± 50.8	0.384
Vfst (mL)	182.0 ± 83.4	187.5 ± 119.5	0.779
MCC (mL)	435.0 ± 140.4	467.5 ± 181.4	0.129
Pdet (cm H_2_O)	10.0 ± 28.5	22.7 ± 14.8	0.016 *
MUCP (cm H_2_O)	56.4 ± 20.9	58.5 ± 21.3	0.517
FUL (cm)	27.6 ± 5.4	27.6 ± 6.6	0.981
UCA (cm^2^ H_2_O)	900.5 ± 386.4	974.3 ± 392.0	0.174

Qmax, maximum flow rate; RU, residual urine; Vfst, bladder volume at first desire to void; MCC, maximum cystometric capacity; Pdet, detrusor pressure at peak flow; MUCP, maximum urethral closure pressure; FUL functional urethral length; UCA, urethral closure pressure area. Values are expressed as mean ± standard deviation or numbers * Statistical significance; Paired *t*-test.

**Table 5 biomedicines-12-02288-t005:** Changes in voiding diaries at baseline and six months after treatment. Data are given as mean ± standard deviation.

n = 34	Baseline	6 Months Post-Treatment	*p* Value *
Frequent urination per 24 h	8.1 ± 2.8	7.2 ± 2.1	0.034 *
Voided urine volume per time (mL)	218.8 ± 93.7	219.8 ± 98.3	0.913
Maximum urine volume (mL)	394.5 ± 146.8	388.7 ± 159.2	0.788
Urge incontinence per 24 h	2.0 ± 1.9	0.9 ± 1.4	0.001 *
Average nocturia per 24 h	0.8 ± 0.8	0.8 ± 0.8	1.0

Values are expressed as mean ± standard deviation or numbers * Statistical significance; Paired *t*-test.

**Table 6 biomedicines-12-02288-t006:** Changes in vaginal and urethral topography at baseline and six months after treatment. Data are given as mean ± standard deviation.

n = 34		Baseline	6 Months Post Treatment	*p* Value *
Bladder neck mobility		1.6 ± 0.3	1.3 ± 0.2	0.003 *
Vaginal width (cm)	Resting	3.1 ± 0.4	2.9 ± 0.3	0.080
	Valsalva	3.5 ± 1.1	2.9 ± 0.8	0.004 *
Vaginal area (cm^2^)	Resting	3.1 ± 0.5	2.8 ± 0.5	0.018 *
	Valsalva	4.0 ± 1.0	3.1 ± 0.9	0.001 *
Proximal urethral rotation angle		15.3 ± 5.0	11.6 ± 3.1	0.009 *

Values are expressed as mean ± standard deviation or numbers * Statistical significance; Paired *t*-test.

## Data Availability

The data that support the findings of this study are available from the corresponding author, C.Y. Long, upon reasonable request.

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
