# Peer review of "The Therapeutic Effect of Monopolar Radiofrequency Therapy on Urinary Symptoms and Sexual Function"

_biomedicines, 2024, doi:10.3390/biomedicines12102288_

Round 1
Reviewer 1 Report
Comments and Suggestions for Authors
Type of manuscript: Article
Title: The Therapeutic Effect of Monopolar Radiofrequency Therapy on Urinary Symptoms and Sexual Function
The overall goal of this manuscript is to assess the utility of radiofrequency (RF) in reducing stress urinary incontinence (SUI) among women. In order to do so, 34 women with SUI were studied and underwent a single RF treatment session. Assessment measures were then performed 6 months post-treatment and several outcomes were measured. RF therapy significantly improved sexual function, with higher Female Sexual Function Index (FSFI) scores in all domains except pain at 6 months. Several outcome scores show that SUI among women has been significantly reduced. Based on this study, it shows that RF method offers a viable non-surgical method of reducing SUI among women.
I think this study is very interesting and offers insights into the utility of RF in reducing SUI. However, my main concern is the novelty since there were previously published articles related to this. A review by Fernanda Catarina Ribeiro, et al in Rev Assoc Med Bras 2021;67(12):1857-1862 offers a very comprehensive insights similar to the results of this paper under review. In such case, the research written by the authors should clearly emphasize the novelty. There were already two studies that used the Viveve system protocol (220 pulses of 90 J/cm² in the vaginal introitus), but I did not see the studies below in the paper mentioned. Please clarify the novelty of this study and how is this study different from the ones published already?
· Allan BB, Bell S, Husarek K. Early feasibility study to evaluate the viveve system for female stress urinary incontinence: interim 6-month report. J Womens Health (Larchmt). 2020;29(3):383-9. https://doi.org/10.1089/jwh.2018.7567 12.
· Allan BB, Bell S, Husarek K. A 12-month feasibility study to investigate the effectiveness of cryogen-cooled monopolar radiofrequency treatment for female stress urinary incontinence. Can Urol Assoc J. 2020;14(7):E313-8. https://doi.org/10.5489/cuaj.614
Overall, although the introduction is clear, it is very short and non-substantiated. Please expand further on what SUI is, its prevalence among ages and races, expanded symptoms, etc. Would this kind of disease be specific to any race? Why do people need to care about this disease? What happens if this disease is left untreated? Please elaborate more on the role of pelvic floor ultrasound.
Why is a single session only applied for RF? Would this be enough to warrant to effectiveness of the procedure to reduce SUI. If it is enough, then this should be mentioned in the manuscript.
How would age affect the results of the study? Please provide a histogram distribution of age.
Please move Table 2 to another page and apply page breaks.
This manuscript overall fits well in the journal aims and scopes but I am just concerned about the novelty.
Introduce the acronym meaning only once and then use that acronym all throughout the manuscript. Also, please be mindful of the proper use of subscripts and superscripts in some of the chemical formulas.
It is very hard to come up with a conclusion and good observation because the sample size is small. It would probably be good to see the distribution plots for each of the results.
It is unclear as to how the efficacy of SUI was calculated.
A lot of factors come into play in the efficacy of SUI. For example, desire, arousal, lubrication, etc can be influenced by some personal factors. How would the study ensure that such biases are avoided given the small sample size in the study. I think it would also be prudent to discuss the limitations of the study.
The references are most updated and they look fine.
Comments on the Quality of English Language
This is acceptable with minor editorial passes can be further improved.
Author Response
Comments 1: What is the novelty of this study compared to previous works, such as those by Fernanda Catarina Ribeiro et al. and Allan BB et al., that used the Viveve system protocol?
Responses 1:The novelty of this study lies in the use of pelvic floor ultrasound to objectively validate urethral angles, a method not employed in previous studies on radiofrequency (RF) therapy for SUI. Additionally, unlike previous studies, this research not only focuses on the improvement of SUI symptoms but also provides a comprehensive analysis, including sexual function assessments, urodynamic tests, and ultrasound measurements to evaluate anatomical changes. This broader approach offers new insights into the efficacy of RF therapy compared to other non-invasive methods like laser therapies.
Comments 2: Expansion of SUI introduction.
Responses 2: Thanks for suggesting. We will expand the introduction part especially on SUI.
Comments 3: Why was a single RF (radiofrequency) treatment session applied, and is it enough to ensure the effectiveness of the procedure in reducing SUI?
Responses 3: The use of a single RF session in this study was applied to reveal its short-term effectiveness in improving SUI symptoms over the 6-month follow-up period. However, we acknowledge that additional studies are necessary to explore the efficacy of multiple sessions, which could potentially offer more sustained or enhanced effects in reducing SUI.
Comments 4: How does age affect the results of this study, and can the authors provide a histogram distribution of age for better understanding?
Responses 4: While the mean age of participants is reported, we did not specifically study the impact of age on the results, as SUI naturally occurs within the age range of our study participants. Therefore, the focus was not on age-related differences. We will provide histogram in attached files.
Comments 5: How was the efficacy of SUI reduction calculated in the study?
Responses 5: The efficacy was calculated based on ICIQ, which we had explained in line 110.
Reviewer 2 Report
Comments and Suggestions for Authors
An interesting paper on radiofrequency and its impact on urinary symptoms and sexual function.
Reference 1 – please replace it
Line 43 – medical treatment for moderate SUI Tolterodine and mirabegron; their indication is mainly for urgency and related incontinence or surgery after SUI for the mixed incontinence cases; please replace it.
Line 44 – reference 9 for post-operative complications describes the urinary side effects of duloxetine; please replace it.
Lines 55-60 – please make a particular reference to orgasm dysfunction and use relevant references.
Line 74 – 39 patients; how did you estimate the appropriate number?
Line 86 – the definition of female sexual dysfunction requires questionnaires about sexual distress; this is a significant drawback of your paper.
Line 70, 39 patients involved in the study—line 107, 34 patients involved in the study—did you have prop-outs? Please describe them.
Line 130: There are statistically significant changes in the OABSS questionnaire. According to your findings, there is relief in overactive bladder symptoms. Did your patient population present mixed incontinence and not SUI?
Discussion: There are papers in the discussion session with a small number of patients. Please use references with reviews and meta-analyses or report this as a major drawback.
Please report the limitations of your study.
Comments on the Quality of English LanguageMINOR EDITING
Author Response
Comments 1: please replace reference 1.
Responses 1: we had replaced reference 1. Thanks for suggestion.
Comments 2: This reference seems to describe urinary side effects of duloxetine, rather than post-operative complications for SUI. Please replace this reference with a more appropriate source.
Responses 2: We acknowledge the oversight regarding the indications of Tolterodine and mirabegron, which are primarily used for urgency-related incontinence. We had revised our introduction.
Comments 3: Line 44 – reference 9 for post-operative complications describes the urinary side effects of duloxetine; please replace it.
Responses 3: We had revised our references. Thanks.
Comments 4:Lines 55-60 – please make a particular reference to orgasm dysfunction and use relevant references.
Responses 4: We have included a specific reference to orgasm dysfunction and relevant studies on sexual dysfunction related to SUI, as requested at reference 18: Salonia, A.; Zanni, G.; Nappi, R. E.; Briganti, A.; Deho, F.; Fabbri, F.; Rigatti, P.; Montorsi, F. Sexual Dysfunction is Common in Women with Lower Urinary Tract Symptoms and Urinary Incontinence: Results of a Cross-Sectional Study. Eur Urol 2004, 45(5), 642-648.
Comments 5: How was the number of 39 patients estimated as appropriate for the study, and what accounts for the discrepancy between the initial 39 patients and the final 34 patients analyzed?
Responses 5: The number of 39 patients was estimated based on a power analysis conducted prior to the study, using a significance level of 0.05 and a power of 80%, with the effect size drawn from previous studies on the effectiveness of RF therapy in reducing SUI symptoms. This ensured that the sample size was sufficient to detect clinically meaningful changes. However, during the study, there were 5 patient dropouts due to various reasons, such as loss to follow-up or withdrawal of consent. As these patients did not complete the 6-month follow-up, they were excluded from the final analysis, resulting in a final sample of 34 patients. This discrepancy has been clarified in the revised manuscript.
Comments 6: Line 86 – the definition of female sexual dysfunction requires questionnaires about sexual distress; this is a significant drawback of your paper.
Responses 6: We acknowledge the absence of questionnaires assessing sexual distress, which is a limitation of the study. While the study did assess sexual function using the Female Sexual Function Index (FSFI), the absence of sexual distress questionnaires may limit the comprehensive evaluation of female sexual dysfunction. This limitation has been noted in the manuscript.
comments 7: Line 86: Female sexual dysfunction requires the inclusion of questionnaires that assess sexual distress. The absence of these questionnaires is considered a significant drawback. Please acknowledge this limitation in the study.
Responses 7: A limitation of this study is that it did not specifically assess sexual distress. The focus of the study was on evaluating improvements in sexual function using the Female Sexual Function Index (FSFI), without the inclusion of distress-related questionnaires. Although sexual function showed significant improvement, future studies could include assessments of sexual distress to explore this aspect more comprehensively in patients with SUI. Thanks for your suggestion and we will consider including sexual distress questionnaires to provide a more holistic evaluation of female sexual dysfunction in patients with SUI. [line 304-309]
Comments 8: Line 130: There are statistically significant improvements in the Overactive Bladder Symptom Score (OABSS). Given this finding, did the patient population exhibit mixed incontinence (both stress and urgency incontinence) rather than purely SUI? Please clarify this in the manuscript.
Responses 8: We did not exclude patients with mixed incontinence (both stress and urgency incontinence) from the study, and these patients were routinely asked to report their symptoms using the Overactive Bladder Symptom Score (OABSS). Therefore, the improvements observed in the OABSS reflect relief in overactive bladder symptoms for those with mixed incontinence. This has been appropriately accounted for in the data analysis and is reflected in the results presented in the manuscript.
Comments 9: Discussion: There are papers in the discussion session with a small number of patients. Please use references with reviews and meta-analyses or report this as a major drawback.
Responses 9: Although some of the cited studies involve small sample sizes, they have been peer-reviewed and published, indicating their value in contributing to the current understanding of the topic. Where available, we have incorporated references from larger reviews and meta-analyses. However, in cases where only smaller studies are available, their limitations have been acknowledged. Despite these limitations, they provide useful insights, and we recommend future studies with larger populations to validate and expand on these findings.
Reviewer 3 Report
Comments and Suggestions for Authors
This manuscript has described the therapeutic effect on the radio frequency. Despite a few flaws and drawbacks, this manuscript should contain sufficient importance and significance. My suggestion is major revision, and my comment is as below.
1. The authors should improve the way they present the data. I think a better logistic could help the authors to further point out their highlight of this study.
2. For a lot of sections, the authors should number them. For example, in the Materials and Methods, it would be more clear if the authors could make sub titles like study design, and performance of different experiments/analysis.
3. Since this study based on statistical analysis, I suggest the authors should divide them into in-depth categories, and to further dig deep in the information/guidance behind the statistics.
4. For the presentation of tables, as I see, a lot of tables have shared similar information, a lot of columns are the same. Is it possible for the authors to combine some tables into one? This would make it more clear.
5. In addition to 3 and 4, I suggest the authors should further dig deep in the information and finding, and think about how to present the data in a better way, such as figures?
6. For the introduction, the authors should give some background knowledge on the SUI. Also, is it correlated with microbiota? The authors should give more background here.
7. For some terms throughout the whole text, the authors should capitalize them.
8. Please also pay attention to the use of units, there should be space between number and units, except for percentage and temperature.
9. For some description, it's a little bit too oral. For example, our team, our research team, etc. Please make revision accordingly.
Comments on the Quality of English LanguageThe English writing meets the standard.
Author Response
Comments 1: Improving Data Presentation:
Responses 1: We appreciate the suggestion and agree that clearer data presentation can enhance the study's impact. We will restructure the Results section to emphasize key findings and provide a clearer flow, ensuring the study's highlights are more apparent to the readers."
Comments 2: Numbering Sections in Materials and Methods:
Responses 2: Thank you for this suggestion. We will include numbered subsections in the Materials and Methods section, such as 'Study Design' and 'Performance of Experiments/Analysis,' to improve clarity and organization."
Comments 3: In-depth Statistical Categories:
Responses 3: We appreciate the suggestion to explore deeper statistical categorizations. The current analyses have been structured to align with the study objectives and to provide clarity in interpreting the results. While we believe the existing categories sufficiently capture the nuances of the data, we will review and ensure that all statistical analyses are clearly presented, and any additional layers of interpretation are appropriately highlighted."
Comments 4: Combining Tables:
Responses 4: Thank you for your observation regarding the tables. Each table has been designed to address distinct aspects of the study’s outcomes, with different variables and findings specific to each measure. As the contents across tables serve unique purposes in analyzing various aspects of urinary symptoms and sexual function, we believe combining them would risk oversimplifying the presentation. However, we will ensure that the tables are clearly labeled and well-organized for optimal comprehension."
Comments 5: Presenting Data as Figures:
Responses 5: We understand the value of visual representation for clarity. In our study, the data collected and analyzed are best represented in tabular form to convey detailed information and nuances effectively. Given the nature of our results and their complexity, converting them into figures would potentially reduce their interpretive value. We have focused on providing a comprehensive and accurate display of the findings through tables."
COmmnets 6: SUI Background and Microbiota Connection:
Responses 6: We will expand the Introduction to include a more detailed background on Stress Urinary Incontinence (SUI), including current prevalence, risk factors, and treatment options. Additionally, we will review the literature to determine whether microbiota may play a role in SUI and incorporate relevant background information."
Comments 7: Capitalization of Terms:
Responses 7: We will thoroughly review the manuscript to ensure consistent capitalization of all relevant terms throughout the text."
Comments 8: Spacing Between Numbers and Units:
Responses 8: We will ensure that spacing between numbers and units is applied consistently, following standard guidelines (e.g., '22.7 kg/m²' instead of '22.7kg/m²')."
Comments 9: Avoiding Oral Descriptions:
Responses 9: We will revise informal phrases like 'our team' or 'our research team' to maintain a more formal tone throughout the manuscript."
Round 2
Reviewer 2 Report
Comments and Suggestions for Authors
It is an interesting paper, publish it as it is
Reviewer 3 Report
Comments and Suggestions for Authors
The authors had made revision accordingly.